# Using Ultrasound Image Augmentation and Ensemble Predictions to Prevent Machine-Learning Model Overfitting

**DOI:** 10.3390/diagnostics13030417

**Published:** 2023-01-23

**Authors:** Eric J. Snider, Sofia I. Hernandez-Torres, Ryan Hennessey

**Affiliations:** U.S. Army Institute of Surgical Research, JBSA Fort Sam Houston, San Antonio, TX 78234, USA

**Keywords:** ultrasound imaging, deep learning, machine learning, artificial intelligence, image classification, shrapnel, medical imaging, overfitting, data augmentation, ensemble predictions

## Abstract

Deep learning predictive models have the potential to simplify and automate medical imaging diagnostics by lowering the skill threshold for image interpretation. However, this requires predictive models that are generalized to handle subject variability as seen clinically. Here, we highlight methods to improve test accuracy of an image classifier model for shrapnel identification using tissue phantom image sets. Using a previously developed image classifier neural network—termed ShrapML—blind test accuracy was less than 70% and was variable depending on the training/test data setup, as determined by a leave one subject out (LOSO) holdout methodology. Introduction of affine transformations for image augmentation or MixUp methodologies to generate additional training sets improved model performance and overall accuracy improved to 75%. Further improvements were made by aggregating predictions across five LOSO holdouts. This was done by bagging confidences or predictions from all LOSOs or the top-3 LOSO confidence models for each image prediction. Top-3 LOSO confidence bagging performed best, with test accuracy improved to greater than 85% accuracy for two different blind tissue phantoms. This was confirmed by gradient-weighted class activation mapping to highlight that the image classifier was tracking shrapnel in the image sets. Overall, data augmentation and ensemble prediction approaches were suitable for creating more generalized predictive models for ultrasound image analysis, a critical step for real-time diagnostic deployment.

## 1. Introduction

Medical imaging-based diagnostics are highly dependent on properly trained and skilled personnel to collect images and interpret them. Unfortunately, in the austere prehospital environment or mass casualty scenarios the skill level of the available healthcare providers will be variable [1]. Military medicine is trying to develop solutions to overcome this challenge. The civilian healthcare system is not exempt from these potential scenarios either so the need to enable less-skilled healthcare providers to make situation-dependent diagnostic decisions spans both military and civilian healthcare systems. Artificial intelligence (AI) has the potential to address this issue if properly implemented. AI models have been used to automate detection for various medical applications, such as COVID-19 [2,3] or tumor identification [4,5], to assist with identification of abnormalities and accelerate intervention. In austere environments, many modern medical imaging tools are not available, thus increasing the challenge of diagnosing and triaging casualties. Equipment size and power constraints limit the options for frontline medical imaging capabilities resulting in ultrasound (US) imaging being the primary means used in these environments [6,7]. Images obtained by US present a unique computer vision problem due to their high level of operator-based variability, higher image noise compared to other medical imaging modalities, and quality control limitations [8]. Regardless of these issues, US’s low power requirements and minimal size (current versions can fit in a pocket) will continue to reinforce it as the go-to source for frontline medical care [9]. By developing toolsets for US imaging that reduce or even remove training requirements for both image collection and interpretation, we can reduce the cognitive burden in these high intensity stressful situations to enable the trained medical personnel to focus on preservation of life that is a critical part of prehospital and battlefield medicine.

A critical challenge for AI neural network models when interpreting medical images is variability due to tissue types, subject variability, injury severity, noise associated with different medical imaging equipment, and operator image collection technique. This requires that the models developed be specific enough to track small changes in images and generalized enough to not be swayed by artifacts present in the medical image [10]. We previously developed a convolutional neural network for identifying shrapnel in ultrasound images—termed ShrapML—as a triage tool to aid in combat casualty care [11]. The intent is to evaluate how ShrapML performs with unseen phantom subject data and the efficacy of deep learning strategies for creating more generalized models. In summary, this work provides the following primary contributions:The evaluation of the effect of data augmentation on creating a more generalized model for shrapnel detection in ultrasound images.Determination of the benefit of bagging multiple predictive models for improving performance with blind test data.With these additions, we develop a more robust AI model framework for tracking shrapnel in ultrasound images which can allow for a more granular triage of casualties based on shrapnel placement in tissue.

### 1.1. Related Work

#### 1.1.1. Automating Shrapnel Detection in Ultrasound Images

Two previous approaches have been taken for automating shrapnel identification in ultrasound images. The first is through the development of neural network models. We have previously developed a convolutional neural network classification model tuned for identifying shrapnel from B-mode US images [11]. Training and test images were collected using a custom-designed tissue phantom as well as swine tissue [12]. While test accuracy exceeded 90% in these studies, this was with test images randomly split from training data as opposed to new test subjects. In another study we compared the ShrapML model against more than 10 conventional neural network models, such as DarkNet-19 and MobileNetV2. MobileNetV2 had the best test accuracy but ShrapML had similar accuracies with much lower inference times [13]. Shrapnel identification has been extended beyond image classification into object detection using a YOLOv3 framework to recognize shrapnel, artery, vein, and nerve fiber features [14]. By identifying each of these neurovascular structures, a triage metric was characterized for measuring the location of shrapnel relative to the nearest neurovascular structure, providing decision support to the end user about the potential risk associated with shrapnel location. The second approach for automating shrapnel identification was through inducing small vibrations in shrapnel embedded in tissue. This was done using an electromagnet, and the vibrations were detected by US doppler signal [15]. This was successful for robotic guidance to remove shrapnel but is only applicable for ferrous shrapnel material types.

#### 1.1.2. Overview of AI Strategies for Creating More Generalized Models

Extensive review articles have been written on various techniques for improving generalization [16,17]; here we focus on three methods: data augmentation, MixUp image augmentation, and leave one subject out (LOSO) ensemble training.

Data augmentation is the use of transformations on images or the addition of artificially generated images in a dataset during the learning phase. Transformations can include such practices as image zooming in or out, shifting the image in any direction, turning the image, recoloring/decoloring, etc. The image transformations should make sense given the dataset and seek to assist the model in looking at the important features. The addition of augmentations during a learning phase has been shown to have the capability to increase model performance and generalizability. [18,19,20,21,22,23,24]. For example, a recent study improved convolutional neural network model accuracy from 84% to 92% by including image normalization and affine transformations to the image set [25]. For ultrasound imaging specifically, augmentation strategies can help in identifying COVID-19 abnormalities in lung tissue [26] or creating a more generalized model for kidney tissue applications [27]. Further details on data augmentation strategies for medical imaging have been detailed in review articles [28,29].

MixUp image augmentation is the combining of two or more images from a dataset by overlaying them on top of each other. Images are combined as a percentage of the total final image. For image data this can be best illustrated by giving the images a complimentary alpha (measurement of transparency) as shown in Figure 1. Training labels can also be combined or assigned based on the percentage of the labelled image representation in the final MixUp image. MixUp, introduced in 2018, is a method that has been shown to be capable of increasing model performance and generalizability [30]. MixUp-style data augmentation has been used to develop more generalized AI models for computed tomography [31], magnetic resonance imaging [32], and ultrasound imaging [33].

Leave one subject out (LOSO) is a version of leave one out (LOO) used in studies where data from one subject is left out of the training process to blind the model to the subject until the testing phase. This LOSO training process is repeated for every training and blind subject test split to evaluate the model’s generalized performance. Bagging is an ensemble machine learning method and is a portmanteau of bootstrapping and aggregating [34]. Bootstrapping is building multiple models from the same cluster of data with different portions of the dataset used as training and validation for each model [35]. Aggregation is the combining of predictions from multiple models in order to produce a combined prediction. Aggregation is used to overcome the weaknesses of a model by the combination of multiple diverse models. Bagging has been shown to generally outperform single classifiers. Both LOSO and Bagging are useful in medical studies where labeled data is expensive and the number of subjects is small [36]. A recent study highlighted how ensemble learning with multiple models for computer tomography COVID-19 image sets was superior to single-model predictions [37]. Bagging has also been used for kidney [38], thyroid [39], and arterial [40] ultrasound imaging applications.

## 2. Materials and Methods

### 2.1. Shrapnel Image Dataseet

A previously developed ultrasound tissue phantom [12] was used as an imaging platform to obtain a shrapnel image database. Briefly, the phantom was primarily comprised of 10% gelatin in a milk–water mixture with agarose fragments non-uniformly distributed in the phantom that generated high variability across each phantom. Ultrasound imaging using a Sonosite Edge (Fujifilm Sonosite, Bothell, WA, USA) was performed underwater, first collecting baseline images of the phantom and then inserting shrapnel of different sizes and shapes at different depths to collect shrapnel-positive images. Images were isolated for each subject to ensure even distribution between the shrapnel and baseline classes. All data was collected as ultrasound clips, from which frames were extracted. All images were cropped and resized using the Image Processing Toolbox from MATLAB (MathWorks, R2021b). Each phantom made was considered a different subject, allowing the allocation of images between testing and training sets before any model training occurred (described below).

### 2.2. Model Architecture Overview of ShrapML and MobileNetv2

Building on previous work [11,13], we used two neural network architectures which had the strongest performance and computational efficiency for detecting shrapnel in ultrasound images. The first model, ShrapML, is an original architecture that has been Bayesian optimized with the phantom image database. ShrapML takes 512 × 512 images as an input and passes them through 6 blocks of 2D convolutional layers with rectified linear unit (ReLU) activators, each followed by a max pooling layer. Then, there is a dropout layer Bayesian optimized to 36.41%, followed by a flattening layer, and two consecutive fully connected layers. Finally, ShrapML has a Softmax layer that feeds the classification output as Baseline or Shrapnel.

MobileNetv2 [41] was the second model evaluated, chosen as a mainstream image classification architecture that has already shown promising results when trained with the shrapnel dataset and compared against more than ten conventional models [13]. Briefly, MobileNetv2 has 53 layers, consisting of 3 convolutional layers and 7 bottleneck modules. The image input layer was modified to accept the 512 × 512 images in the shrapnel dataset as input, and the model output layer was adapted to a binary Baseline or Shrapnel prediction.

### 2.3. Preprocessing Training Data

We used two data augmentation approaches for training data. First, conventional affine transformation methods were used which included: random rotation from 0 to 360 degrees, random scaling between 0.8 and 1.2, random X-plane reflections, and random Y-plane reflections. Second, the blending image augmentations, or MixUp, were used. Baseline images were combined with other baseline images to expand the baseline dataset. Shrapnel images were combined with baseline images create additional shrapnel images with image noise from baseline images. Images resulting from MixUp were kept in folders identifying the subject of origin of both images, to avoid crossover between subjects. For example, if a baseline image from subject *a* was combined with a shrapnel image from subject *c*, the resulting image would be used for training a blind model that is then tested on subject *b*.

### 2.4. Blind Subject Model Training

Data for five training subjects were split between training (200 images for each category) and testing (70 images for each category) so that all splits had an equal number of images. Each of the five subjects was comprised of multiple phantoms but there was no spillover between each subject. The algorithm training process is illustrated in Figure 2 and described below.

The training sets were arranged into five leave one subject out (LOSO) holdouts making one of the five subjects a blind test for each trained model. Training was performed for each LOSO incorporating the different augmentation approaches, which rendered four training groups: (1) no augmentation, (2) only affine transformations, (3) only MixUp, and (4) both affine and MixUp augmentations. The same total number of images was used for each of the training groups (800 total images for each category), and the images were randomly selected from the folders. ShrapML and MobileNetv2 neural networks were both trained with all four groups for a maximum of 100 epochs, with a 0.001 learning rate, mini batch size of 28, and RMSprop optimizer. The network was allowed to end training when the minimum validation loss had not decreased for 10 consecutive training iterations, and the epoch with the lowest validation loss was selected for subsequent performance evaluation. This was replicated three times for each LOSO.

Next, bagging (bootstrap aggregating) was implemented to improve blind test performance, using the ShrapML model replicate with the best recall score for each LOSO trained. Aggregated models were evaluated using absolute categorical predictions or summing up confidences across the five LOSO holdouts and then determining the categorical predictions based on the total confidence value. For a different ensemble approach, the two lowest confidence score models for each image prediction were removed from the aggregation process for both the categorical and confidence methodologies. All testing for the LOSO bagging models was performed on test sets from two different subjects (6 & 7) that were not part of the previous training or test sets. In other words, there were a total of 7 subjects, 5 used for the training and validation and 2 others held out exclusively for the ensemble testing.

Finally, confidence scores from all the LOSO models were considered and only the three models with the highest confidence in their predictions were chosen as a top 3 ensemble. These results were compared for all four training groups to select which training group showed the best performance. This has been termed LOSO bagging as LOSO was used as the method for bootstrapping and then the produced models were aggregated (Figure 2).

### 2.5. Evaluating Model Performance

After all models were trained, predictions for the held out subject test sets were scored using confusion matrices, accuracy, F1 scores, precision, recall (sensitivity), specificity, and area under the ROC curve (AUROC). Of all performance metrics, recall was used as the distinguishing metric to select the best model, as it indicates how well the model identifies positive instances, or the true positive rate. A higher recall score also means fewer instances of false negative predictions, the preferred bias for shrapnel detection. Models trained with a higher recall score are more likely to predict shrapnel in a shrapnel-positive image, and less likely to predict a shrapnel-positive image incorrectly (a false negative). Performance metrics were found for triplicate models and were shown as average values throughout.

To illustrate the area of the image that the models focused on when making a prediction, gradient-weighted class activation mapping (GradCAM) was used to generate heatmap overlays [42]. This analysis was performed for each LOSO model and test image when applicable using MATLAB, and heatmap overlays were saved with their respective ground truth and prediction labels, confidences. For each model’s results section, representative images are shown with the GradCAM overlays highlighting the region where the classification prediction was based on.

## 3. Results

### 3.1. Initial Performance of ShrapML during Blind Subject Testing

We have previously developed an image classifier neural network model for shrapnel detection (ShrapML) in ultrasound images with greater than 95% accuracy in split test image sets. Here, we evaluate its effectiveness in new tissue phantom subjects prior to introducing data augmentation and other approaches to improve test performance. Overall, the model had a high false negative rate, resulting in a specificity of 0.519 (Figure 3). Accuracy was 0.688 and the area under ROC curve held around 0.772 across three replicate training runs. The performance challenges were further shown through GradCAM overlays to explain what the model was tracking for its prediction across the different prediction categories (Figure 4). When the trained model was correct in its prediction, it identified shrapnel successfully or did not identify artifacts in baseline images. However, it was evident that shrapnel identification was tracking too many features to focus in on shrapnel in false negatives. For false positives, non-shrapnel tissue artifacts were identified as shrapnel.

### 3.2. Effect of Data Augmentation and Model Architechture on Improving Blind Subject Test Performance

Initial training datasets were comprised of five tissue phantom subjects. These training sets were configured into five leave one subject out (LOSO) training sets to make one of the five phantoms a blind validation subject for each training iteration. Using this methodology, the performance variability was evident for ShrapML, as accuracy varied from 0.688 to 0.755 (Table 1). More pronounced variability was evident with recall (0.538 to 0.857) and specificity (0.519 to 0.900), indicating various LOSO holdouts varied greatly in false positive and negative rate. These LOSO holdouts were used for the remainder of the study to better assess performance consistency for different blind test subjects.

To improve ShrapML performance, two augmentation methodologies were utilized, and four training groups with different augmentation combinations were used to train the model. For all training groups, prediction results were similar, but the affine transformations only group had a slight reduction in its false positive rate (Figure 5). For further comparison, the same training setup was used with MobileNetV2 instead of ShrapML, as we have previously shown it performed best for shrapnel identification in ultrasound image data sets [13]. Overall, prediction results were similar across the MobileNetV2 models, except for MixUp augmentation only, which had a reduced false positive rate (Figure 6).

For a more objective comparison of each model’s performance, a summary of performance metrics is shown in Table 2. All ShrapML architecture models had a higher accuracy when compared to MobileNetV2 trained ones, with affine augmentations only having the strongest performance at 0.749. Similarly, affine augmentations without MixUp had the strongest performance for precision, specificity, and AUROC (Table 2). Recall and F1 score were strongest with MixUp augmentation only ShrapML trained models. In addition, the affine and MixUp augmentation group performed well for ShrapML, however, it had the worst overall performance for the MobileNetV2 architecture. Overall, ShrapML models outperformed MobileNetV2 with these augmentation strategies. Affine transformation augmentation without MixUp performed the best for most performance metrics, however, blind test accuracy was still below 75%.

### 3.3. Evaluation of Ensemble Prediction Pooling on Blind Subject Test Performance

Next, LOSO bagging was evaluated to improve blind test performance. The highest recall performance metric model of the three replicate models was selected for each LOSO holdout. Two additional phantoms which were not included in the training/validation sets were used for blind testing for this use case. The aggregation was evaluated using absolute categorical—shrapnel or baseline—predictions or summing up confidences across the five LOSO holdouts. In addition, ensemble predictions were evaluated when only considering the top three most confident model predictions for each image—termed Top 3 LOSO. Predictions had a higher true positive and negative rate after aggregation, and overall performance across each aggregation approach were very similar (Figure 7). Performance metrics reached as high as 0.85 accuracy with the Top 3 confidence-based LOSO bagging method (Table 3). This methodology outperformed all other approaches except for precision being 0.01 higher for the Top 3 LOSO categorical-based LOSO bagging method.

The Top 3 confidence-based LOSO bagging method was used for the previous four ShrapML data augmentation strategies to confirm that affine augmentation only remained the best performing. Across test predictions, this augmentation strategy training group had the most similar false positive and negative rate, while other models were slightly biased toward false positives (Figure 8). Without data augmentation, the model accuracy with ensembles was 0.764, while each augmentation strategy model scored above 0.825 highlighting how data augmentation aided in making ShrapML more generalized for this application (Table 4). Affine augmentation only and both affine and MixUp augmentation groups had the strongest performance metrics with both surpassing 85% blind testing accuracy.

As the affine transformation and MixUp training group performed slightly better than the other strategies, GradCAM overlays were created to highlight how the various LOSO holdouts performed for different prediction categories (Figure 9). In general, there was strong GradCAM agreement across the LOSOs when accurate, but false predictions were less consistent with different models recognizing different artifacts in the tissue phantoms.

## 4. Discussion

Machine learning models can simplify and automate interpretation of medical imaging to aid the clinical decision-making process in generating a diagnosis when the models are designed to handle subject variability observed clinically. In this work, we focused on shrapnel detection in ultrasound images and worked towards generalizable models for handling new subjects not initially included in the training sets. While more clinical data is always the first answer for solving overfitting problems, oftentimes the datasets do not exist and, in the cases of rare abnormalities, it will be very challenging to collect enough data to adequately account for subject variability. Instead, various machine learning training and prediction techniques can be used to augment images or create a more robust prediction process.

We first highlighted data augmentation methods. Affine transformations have long been used for preventing overfitting training data and there are an infinite number of ways these operations can be configured. Here, we focused on x/y reflections, rotation, and zoom transformation types, each applied at a random magnitude to training images. Slight improvements were noticed in blind test data with just this augmentation approach, increasing overall accuracy from 0.724 to 0.749. Data augmentation can be further optimized for this application by Bayesian optimization of the magnitudes and type of affine transformation to further improve testing accuracy. Another augmentation approach used in this work was MixUp. This approach is ideal for ultrasound imaging applications such as this; shrapnel could be added into baseline images to create additional training images with all the noise and signal artifacts of baseline images. With MixUp alone, accuracy was only slightly improved as well, increasing from 0.724 to 0.731. This is evidence of moving towards a more generalized model. A model is considered generalizable when the model begins to only have slight changes in accuracy when tested on novel datasets. Data augmentation lowered training accuracy with all this additional noise introduced in the training sets, but this resulted in a more consistent performance across the various blind test runs.

A more generalized model by itself could not bring prediction accuracies above 75%, so we explored the use of ensemble prediction techniques. Often when making a diagnosis or recommending a medical treatment, a medical consensus is relied upon rather than a single medical opinion. This is paralleled by the ensemble bagging approach used, as each of five models make their prediction and the category with the majority consensus is chosen. Overall, all the ensemble methods evaluated were able to improve the prediction accuracy. We modified the ensemble approach to only rely on the three most confident predictions. If a model was guessing (i.e., low confidence), it seemed advantageous to remove these predictions from the aggregate score. This approach further improved training accuracy to more than 85%. However, it was still evident in the GradCAM overlays that when the ensemble was inaccurate, one or two or the models was still accurate at finding or not finding shrapnel, suggesting improvements to the ensemble bagging approach can further refine the prediction accuracy. One approach would be to not only pool model predictions but also pool multiple image frames so that the diagnosis is informed by a sequence of images rather than by a single frame. Pooling multiple images would be trivial for ultrasound imaging, where video clips can be recorded at a high frame rate. Another approach would be to refine how confidences were ensembled, by including the concentrations observed in GradCAM overlays with how they coincide with other model predictions to determine which models agree and which should be excluded for the current prediction. Regardless, the ensemble approach as presented in this work improved accuracy from an initial 68% test accuracy to over 85%.

The next steps for this research effort will take three main directions. First will be extension from image classification toward object detection and semantic segmentation models as these approaches have more diagnostic potential for medical imaging. We have previously shown that a YOLOv3 object detector can successfully detect shrapnel [14] but similar augmentation ensemble methods may be required to make models more generalized. Segmentation models have the potential to track precise edges of foreign bodies in tissue which may be critical for aiding surgical intervention [43]. Second, these more generalized models will be modified for use with ultrasound video clips to improve performance and for more real-time integration with ultrasound equipment. Third, extension of this shrapnel application to animal and human image sets will be pursued as will its use in other areas of the body. Image classification of shrapnel in the chest abdomen can be paired with ongoing research for the extended focused assessment with sonography in trauma (eFAST) exam, a widely used procedure for detecting free fluid in the chest or abdomen in emergency or military medicine applications [44,45,46].

## 5. Conclusions

In conclusion, machine learning techniques can automate medical image interpretation if models are developed which can account for the large subject variability prevalent in medical imaging. Here, we demonstrate how data augmentation can add noise and variability to training sets for reducing overfit of training data sets and improving blind subject accuracy. This was further improved by use of ensemble prediction techniques which pooled predictions from five separately trained machine learning models to improve blind accuracy for shrapnel identification in ultrasound images from 68% to over 85%. Through further refinement of these methods and integration into ultrasound imaging equipment, more generalized models can be created to automate medical imaging interpretation to facilitate the determination of the medical diagnosis.

## Figures and Tables

**Figure 1 diagnostics-13-00417-f001:**
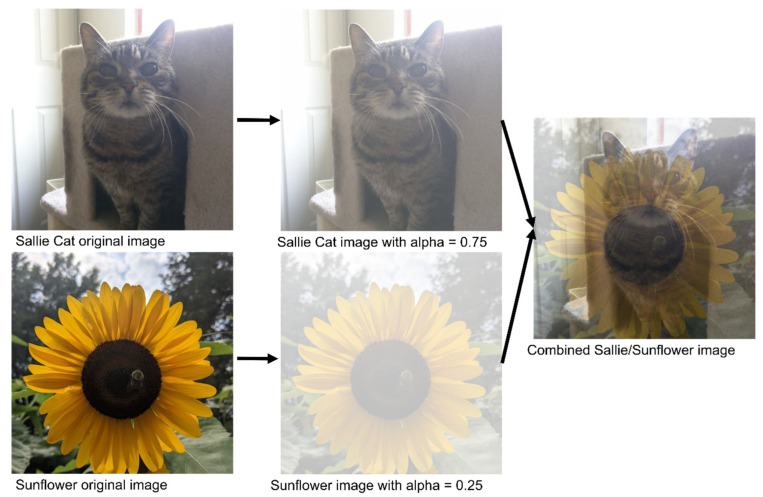
Visualization of the steps of MixUp data augmentation. Given two images to be mixed, the first image (top) is assigned a percentage representation value (between 0 and 1), and the second image is then assigned the complementary percentage representation. The two images are then combined to form the final image.

**Figure 2 diagnostics-13-00417-f002:**
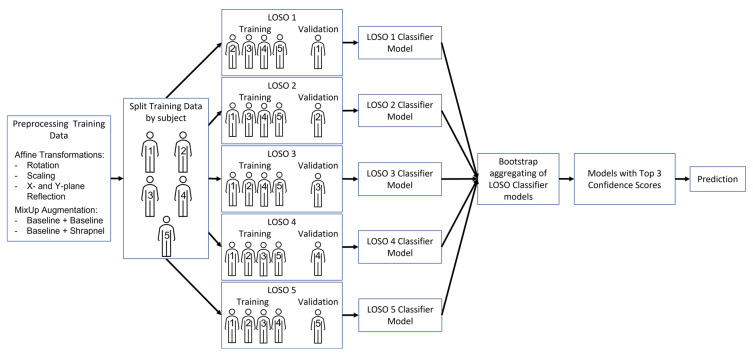
Flowchart describing the methods of this study, from data augmentation to final top 3 predictions, using LOSO training methodology to develop the final ensemble model.

**Figure 3 diagnostics-13-00417-f003:**
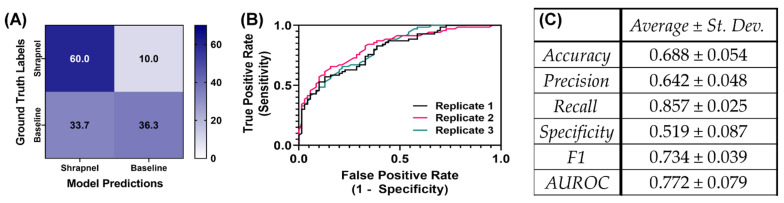
Performance of ShrapML model with blind subject testing. (**A**) Average confusion matrix results across three training replicates for 70 shrapnel and 70 baseline images for a blind test subject. (**B**) Receiver operating characteristic curves and (**C**) summary of performance metrics for three replicate training runs. Results are shown as averages and standard deviations across three replicate trained models.

**Figure 4 diagnostics-13-00417-f004:**
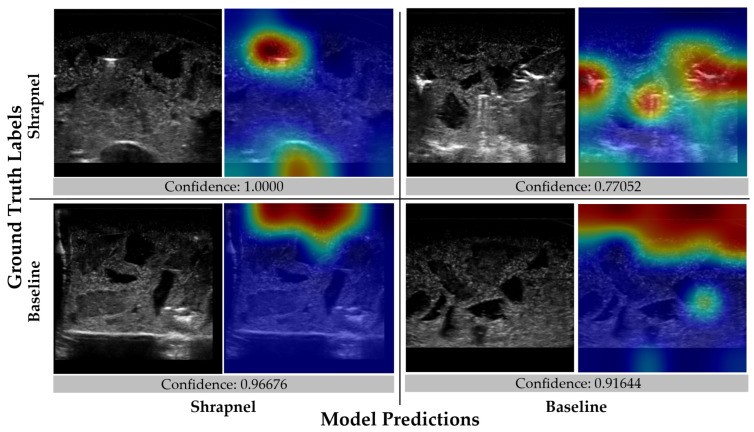
Gradient-weighted class activations mapping overlays for the initial ShrapML model. Representative images are shown with and without GradCAM overlays in a confusion matrix layout along with the prediction confidence.

**Figure 5 diagnostics-13-00417-f005:**
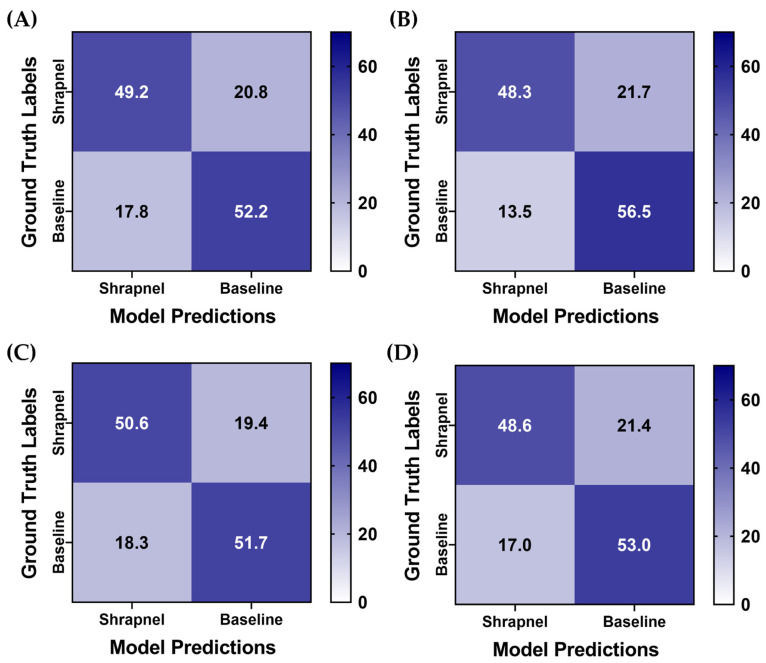
Data augmentation effects on blind test performance for ShrapML. Average confusion matrix results are shown for (**A**) No augmentation, (**B**) Affine augmentation only, (**C**) MixUp augmentation only, (**D**) Both affine and MixUp augmentations. Results are averaged for three training replicates for five LOSO holdout runs. A total of 70 shrapnel and 70 baseline images were used for each blind test run.

**Figure 6 diagnostics-13-00417-f006:**
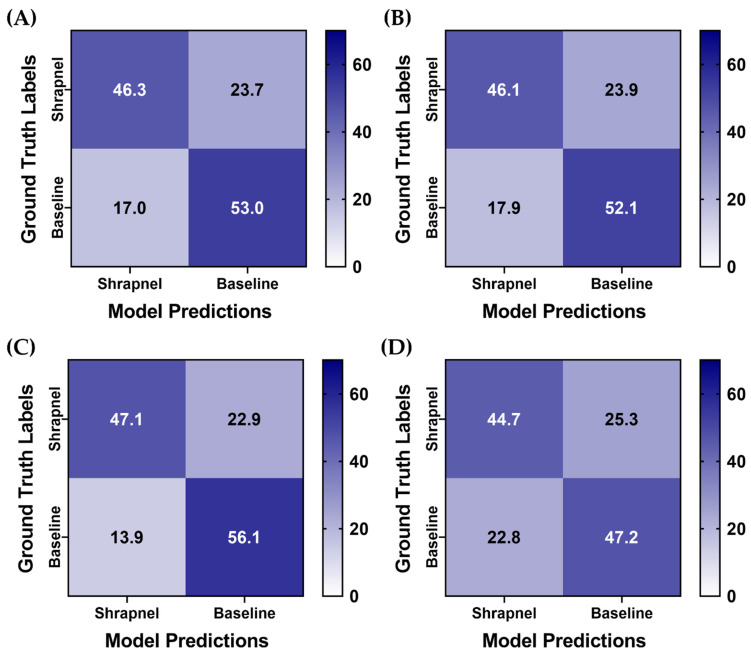
Data augmentation effects on blind test performance with a MobileNetV2 model architechture. Average confusion matrix results are shown for (**A**) no augmentation, (**B**) affine augmentation only, (**C**) MixUp augmentation only, (**D**) both affine and MixUp augmentations. Results are averaged for three training replicates for the five LOSO holdout runs. A total of 70 shrapnel and 70 baseline images were used for each blind test run.

**Figure 7 diagnostics-13-00417-f007:**
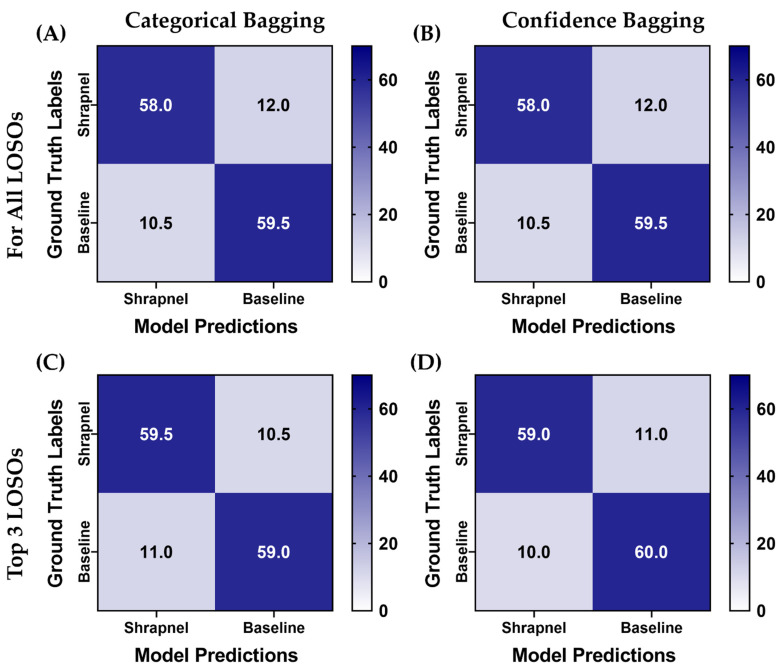
Blind testing performance for affine augmentation-only ShrapML trained models using various ensemble prediction bagging approaches. Average confusion matrices are shown for two blind test image sets for (**A**) categorical and (**B**) confidence ensemble bagging for all five LOSO holdout models. (**C**) Categorical and (**D**) confidence ensemble bagging using the Top 3 confidence models for each prediction. A total of 70 shrapnel and baseline images were used for each blind test run.

**Figure 8 diagnostics-13-00417-f008:**
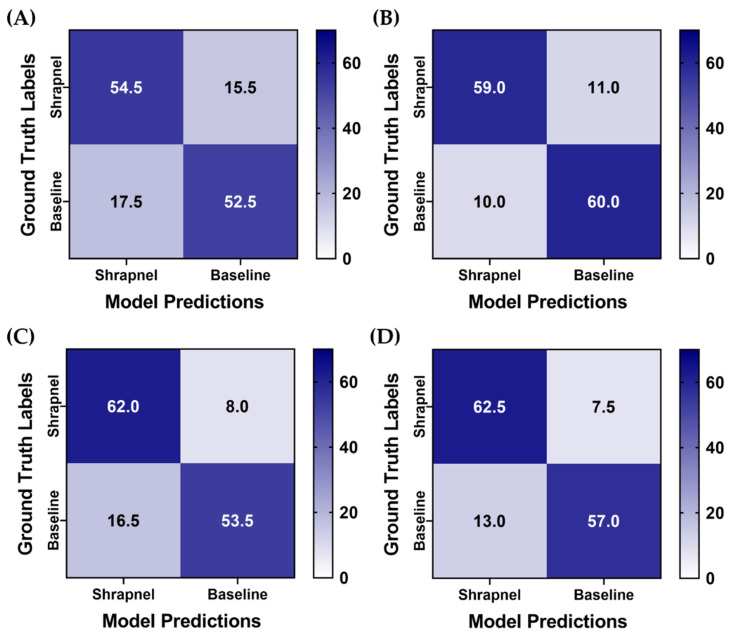
Blind testing performance for Top 3 categorical ensemble bagging across four different data augmentation strategies for ShrapML. Average confusion matrices are shown for two blind test image sets for (**A**) no augmentation, (**B**) affine augmentation only, (**C**) MixUp augmentation only, (**D**) both affine and MixUp augmentations. A total of 70 shrapnel and 70 baseline images were used for each blind test run.

**Figure 9 diagnostics-13-00417-f009:**
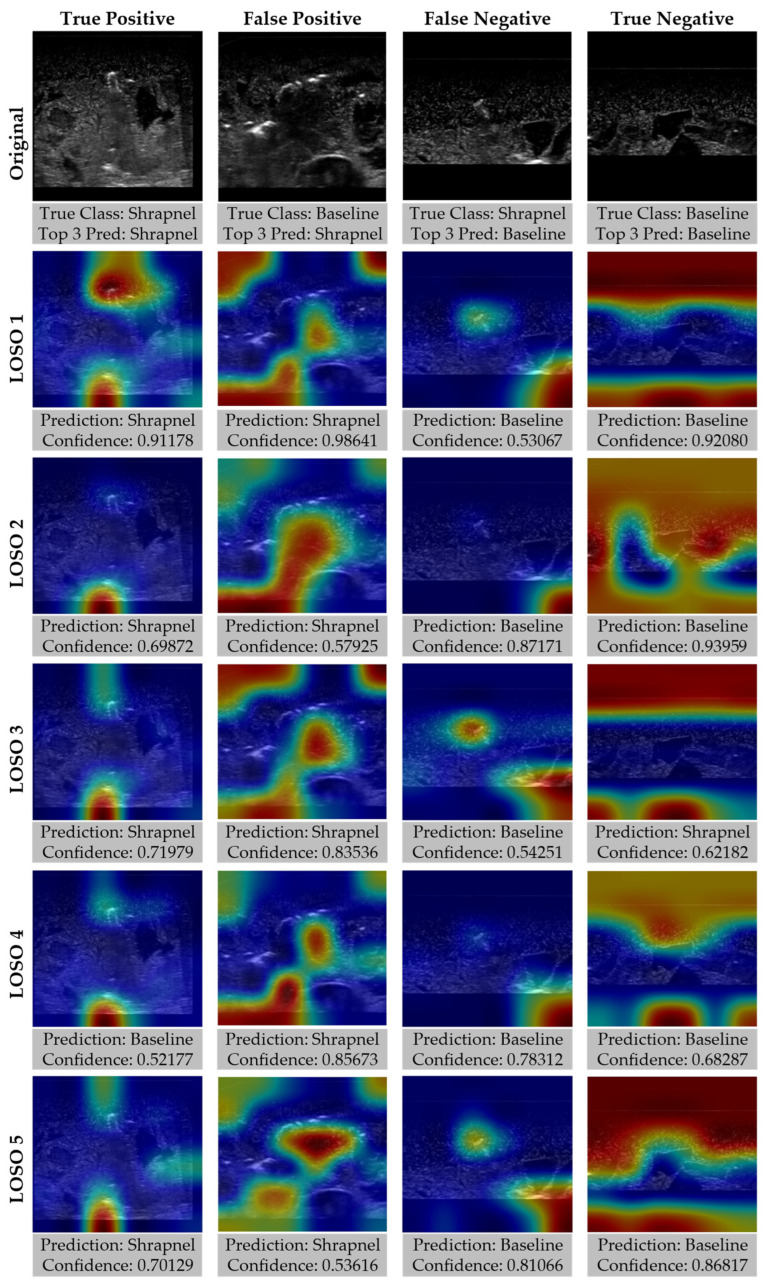
Gradient-weighted class activations mapping overlays for each LOSO holdout for ShrapML model including affine transformations and MixUp data augmentation. Representative images for true positive, true negative, false positive, and false negative prediction categories are shown along with GradCAM overlays for each LOSO, with prediction and confidences.

**Table 1 diagnostics-13-00417-t001:** Effect of leave one subject out (LOSO) holdouts on ShrapML testing performance. Results are shown as average results for n = 3 replicate trained models. Heat map overlay is set between the minimum and maximum value for each performance metric.

	LOSO 1	LOSO 2	LOSO 3	LOSO 4	LOSO 5
Accuracy	0.688	0.757	0.719	0.755	0.702
Precision	0.642	0.794	0.855	0.778	0.694
Recall	0.857	0.695	0.538	0.710	0.714
Specificity	0.519	0.819	0.900	0.800	0.690
F1	0.734	0.740	0.652	0.738	0.702
AUROC	0.772	0.813	0.705	0.831	0.756

heat map overlay is set between the highest value (green) and lowest value (white) for each performance metric.

**Table 2 diagnostics-13-00417-t002:** Summary of performance metrics for different data augmentation strategies with ShrapML and MobileNetV2 models. Average results across three training replicates for each of five LOSO holdout runs are shown. Heat map overlay is set between the minimum and maximum value for each performance metric.

	ShrapML				MobileNetV2			
	No Augmentation	Affine Only	MixUp Only	Affine and MixUp	No Augmentation	Affine Only	MixUp Only	Affine and MixUp
Accuracy	0.724	0.749	0.731	0.726	0.709	0.701	0.737	0.656
Precision	0.753	0.835	0.763	0.824	0.745	0.815	0.802	0.763
Recall	0.703	0.690	0.723	0.694	0.661	0.659	0.673	0.638
Specificity	0.746	0.808	0.739	0.757	0.757	0.744	0.801	0.674
F1	0.713	0.726	0.730	0.695	0.631	0.623	0.678	0.636
AUROC	0.775	0.867	0.794	0.856	0.795	0.859	0.835	0.836

heat map overlay is set between the highest value (green) and lowest value (white) for each performance metric.

**Table 3 diagnostics-13-00417-t003:** Performance metrics for each ensemble method with augmentation + no MixUp ShrapML model architecture. Mean results for two separate blind test subjects are presented. Heat map overlay is set between the minimum and maximum value for each performance metric.

	All LOSOs Categorical Bagging	Top 3 LOSO Categorical Bagging	All LOSOs Confidence Bagging	Top 3 LOSO Confidence Bagging
Accuracy	0.839	0.846	0.839	0.850
Precision	0.854	0.850	0.851	0.860
Recall	0.829	0.850	0.829	0.843
Specificity	0.850	0.843	0.850	0.857
F1	0.838	0.847	0.837	0.848

heat map overlay is set between the highest value (green) and lowest value (white) for each performance metric.

**Table 4 diagnostics-13-00417-t004:** Performance metrics for Top 3 categorical ensemble bagging across four different data augmentation strategies for ShrapML. Mean results for two separate blind test subjects are presented. Heat map overlay is set between the minimum and maximum value for each performance metric.

	No Augmentations	Only Affine Augmentations	Only MixUp Augmentations	Both Augmentations
Accuracy	0.764	0.850	0.825	0.854
Precision	0.764	0.860	0.795	0.828
Recall	0.779	0.843	0.886	0.893
Specificity	0.750	0.857	0.764	0.814
F1	0.769	0.848	0.837	0.859

heat map overlay is set between the highest value (green) and lowest value (white) for each performance metric.

## Data Availability

The datasets generated during and/or analyzed during the current study are available from the corresponding author upon reasonable request.

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
