# Peer review of "Using Ultrasound Image Augmentation and Ensemble Predictions to Prevent Machine-Learning Model Overfitting"

_diagnostics, 2023, doi:10.3390/diagnostics13030417_

Round 1

Reviewer 1 Report

In this manuscript, authors have used ultrasound image augmentation and ensemble predictions to prevent machine-learning model overfitting. This manuscript is well written but requires some significant improvements mentioned below. 

1) Include 3-4 contributions/novelties in the form of bullets after the introduction section.

2) Include a flowchart in the introduction section that should give the complete overview of your study. 

3) Include a related work section along with the comparison table that should highlight the strengths and weaknesses of the previous as well as the proposed method.

Author Response

Reviewer 1

In this manuscript, authors have used ultrasound image augmentation and ensemble predictions to prevent machine-learning model overfitting. This manuscript is well written but requires some significant improvements mentioned below.

Include 3-4 contributions/novelties in the form of bullets after the introduction section.

 We appreciate the thorough review of our manuscript, we have added a bullet point list of 3 – 4  novelties as suggested in the introduction, on line 66.

Include a flowchart in the introduction section that should give the complete overview of your study.

We have modified the figure 2 flow chart to cover a complete overview of the study. However, we have included this as a method figure as opposed to an introductory figure. The figure begins at line 220.

Include a related work section along with the comparison table that should highlight the strengths and weaknesses of the previous as well as the proposed method.

We have expanded on the current related work section discussing different augmentation approaches as suggested, this begins at line 73. This was done in paragraph form as opposed to a comparison table as it flowed better with the current paper structure.

Reviewer 2 Report

The paper presents an interesting idea but some parts of it should be modified:

- The state of the art related to what is proposed needs to be better investigated;

- Figure 2 shows the proposed framework. It should be better integrated into the text;

- The figures need to be improved in terms of quality;

- Section 2.1 describes the dataset used. It should be moved to the experimental part and summarized in a table;

- In the specific context, the following paper should be cited:

Manzo, Mario, and Simone Pellino. "Fighting Together against the Pandemic: Learning Multiple Models on Tomography Images for COVID-19 Diagnosis." AI 2.2 (2021): 261.

Author Response

Reviewer 2

The paper presents an interesting idea but some parts of it should be modified:

The state of the art related to what is proposed needs to be better investigated;

Great suggestion and thanks for reviewing our manuscript. We have previously investigated comparing the ShrapML algorithm to more than 10 state of the art deep learning model architectures. We have more explicitly mentioned this prior study in the introduction and methodological sections. This is on line 75 and 165.

Figure 2 shows the proposed framework. It should be better integrated into the text;

We agree with the reviewer and have expanded on discussing the LOSO method and the figure in the text after first mention. This begins on line 185.        

The figures need to be improved in terms of quality

We have revamped figures 1 and 2 that were most affected from low image quality. The rest of the figures are high resolution, we believe. However, we can further refine the images quality during the proofing process.

Section 2.1 describes the dataset used. It should be moved to the experimental part and summarized in a table

All the datasets used in this study are retrospective. A summary of the number of images used in this study is on line 183. We have left this section in the methods as we believe it fits better there with the flow of the paper. We have modified the methods to further clarify that these images were retrospective not experimental to this study.

In the specific context, the following paper should be cited:

Manzo, Mario, and Simone Pellino. "Fighting Together against the Pandemic: Learning Multiple Models on Tomography Images for COVID-19 Diagnosis." AI 2.2 (2021): 261.

We have added this paper to the related works section

Round 2

Reviewer 1 Report

Most of my comments are addressed. I recommend acceptance of this manuscript. 

Reviewer 2 Report

As far as I'm concerned, there are no further changes to be made